# Experimental Autoimmune Encephalomyelitis of Mice: IgGs from the Sera of Mice Hydrolyze miRNAs

**DOI:** 10.3390/ijms24054433

**Published:** 2023-02-23

**Authors:** Georgy A. Nevinsky, Andrey E. Urusov, Kseniya S. Aulova, Evgeny A. Ermakov

**Affiliations:** Institute of Chemical Biology, Fundamental Medicine of the Siberian Division of Russian Academy of Sciences, Lavrentiev Ave. 8, Novosibirsk 630090, Russia

**Keywords:** C57BL/6 mice, EAE model of human multiple sclerosis, immunization with MOG, catalytic antibodies, hydrolysis of RNAs and micro-RNAs

## Abstract

It was shown that the spontaneous development of experimental encephalomyelitis (EAE) in C57BL/6 mice occurs due to changes in the profile of bone marrow stem cells differentiation. This leads to the appearance of lymphocytes producing antibodies-abzymes that hydrolyze DNA, myelin basic protein (MBP), and histones. The activity of abzymes in the hydrolysis of these auto-antigens slowly but constantly increases during the spontaneous development of EAE. Treatment of mice with myelin oligodendrocyte glycoprotein (MOG) leads to a sharp increase in the activity of these abzymes with their maximum at 20 days (acute phase) after immunization. In this work, we analyzed changes in the activity of IgG-abzymes hydrolyzing (pA)_23_, (pC)_23_, (pU)_23_, and six miRNAs (miR-9-5p, miR-219a-5p, miR-326, miR-155-5p, miR-21-3p, and miR-146a-3p) before and after mice immunization with MOG. Unlike abzymes hydrolyzing DNA, MBP, and histones, the spontaneous development of EAE leads not to an increase but to a permanent decrease of IgGs activity of hydrolysis of RNA-substrates. Treatment of mice with MOG resulted in a sharp but transient increase in the activity of antibodies by day 7 (onset of the disease), followed by a sharp decrease in activity 20–40 days after immunization. A significant difference in the production of abzymes against DNA, MBP, and histones before and after mice immunization with MOG with those against RNAs may be since the expression of many miRNAs decreased with age. This can lead to a decrease in the production of antibodies and abzymes that hydrolyze miRNAs with age mice.

## 1. Introduction

Multiple sclerosis (MS) is a chronic autoimmune disease, the pathogenesis of which is characterized by the demyelination (plaques) of the gray and white matter of the brain and spinal cord, leading to neurodegeneration and brain atrophy [1,2]. The etiology of multiple sclerosis is still unclear; the most accepted theory of pathogenesis assigns the central role to the destruction of myelin-proteolipid shell axons resulting in inflammation bound with important autoimmune reactions ([3] and references therein).

Natural abzymes (ABZs) splitting various oligosaccharides, lipids, peptides, proteins, DNAs, and RNAs, were revealed in the blood of patients with some autoimmune diseases (AIDs) and viral pathologies [4,5,6,7,8,9,10]. ABZs with insignificant activities splitting thyroglobulin [10], polysaccharides [11], and vasoactive neuropeptide [12] were revealed in the sera of some conditionally healthy volunteers. However, the blood of healthy people usually does not contain abzymes [4,5,6,7,8,9,10,13]. Similar to systemic lupus erythematosus (SLE) [9], the blood of MS patients contains abzymes hydrolyzing DNAs and RNAs [13,14,15,16], myelin basic protein (MBP) [17,18,19,20], histones [21], and oligosaccharides [12]. Relative activities (RAs) of IgGs from the cerebrospinal fluids degrading polysaccharides, MBP, and DNAs are, on average, from 35 to 60 times higher than those from the blood of the same MS patients [22,23,24].

MS is a multifactorial disease, the pathogenesis of which could depend on many various factors [25]. Micro-RNAs are small (22–25 nucleotides) non-coding RNAs participating in the post-transcriptional regulation of many genes [26,27], including transcription and neuroinflammation [26,27,28]. In MS and SLE, specific miRNAs characterized by increased expression are revealed in the cerebrospinal fluid and blood [28,29,30,31]. The extracellular miRNAs participate in signaling between cells and regulating neurogenesis, angiogenesis, and cell proliferation [32]. The change in miRNA expression in many cases is associated with pathological processes. As a result of inflammatory processes in SLE and MS, specific miRNAs’ processing, transcription, or maturation can be changed. Some miRNAs may be biomarkers of AIDs [33]. Thus, eliciting additional essential factors of MS pathogenesis and a possible role of miRNAs may be important. It is possible that not only miRNAs but also antibodies and abzymes against miRNA can also have a specific role in the pathogenesis of MS.

The level of abzymes with different activities in patients with MS and other AIDs varies significantly from patient to patient [4,5,6,7,8,9,10]. However, it is difficult to accurately assess which factors, in the case of patients with AIDs, are the main ones in the development of pathologies.

Several experimental autoimmune encephalomyelitis (EAE) mice models well mimic a specific facet of human MS are known (for review, see [34,35]). Analysis of the patterns of development of EAE in the case of two models of EAE-prone mice (C57BL/6 [36,37,38,39] and Th [40,41]) and one model of mice prone to systemic lupus erythematosus (MRL-lpr/lpr [42,43,44]) made it possible to identify several common factors essential for the development of AIDs. It was shown that the spontaneous and antigen-induced development of EAE [36,37,38,39,40,41] and SLE [42,43,44] occurs first of all due to specific changes in differentiation profiles of bone marrow hematopoietic stem cells (HSCs) associated with an increase in lymphocyte proliferation and apoptosis repression in different organs of these mice [36,37,38,39,40,41,42,43,44,45]. Changes in differentiation profiles in EAE and SLE mice during the development of these pathologies are very similar [36]. These changes in EAE-prone mice lead to the appearance of B lymphocytes producing catalytically active antibodies that hydrolyze DNA, MBP, mouse myelin oligodendrocyte glycoprotein peptide (MOG), and five histones (H1-H4) [36,37,38,39,40,41]. In the case of SLE-prone MRL-lpr/lpr mice, during the process of spontaneous and DNA-induced development of SLE was shown appearance in the blood of abzymes hydrolyzing DNA, ATP, and oligosaccharides [42,43,44].

Anti-DNA antibodies (Abs) in SLE, MS, and other AIDs are usually directed against nucleosomal histone-DNA complexes [45]. ABZs with DNA-hydrolyzing activity are cytotoxic: they penetrate the nucleus of the cells and split nuclear DNA leading to cell apoptosis [46,47], which increases concentrations of DNA and its complexes with histones in the blood and acceleration AIDs development [7,8,9,10]. In MS, Abs with MBP-hydrolyzing activity can attack MBP of the myelin-proteolipid sheath of the nerve tissue membranes, leading to a disruption of nerve impulses and providing a harmful role of such ABZs in MS pathogenesis [5].

Abzymes with micro-RNA hydrolyzing activity are found in the blood of patients with MS [48], SLE [49], and schizophrenia [50,51]. Several miRNAs regulate neuroinflammation and are characterized by impaired expression in MS [28]. Antibodies from the blood of MS patients efficiently hydrolyze these miRNAs [48]. A comparison of the development of MS in humans and EAE mice indicates that these processes are substantially similar. Therefore, it was expected that abzymes from the blood of C57BL/6 mice could hydrolyze miRNAs similar to IgGs of patients with MS.

It was shown that C57BL/6 mice are characterized by a very slow spontaneous and MOG-induced development of EAE [33,34,35,36,37,38,39,40,41]. Some typical indicators of EAE development (optic neuritis and other clinical or histological evidence) appear in C57BL/6 mice only 1–2 years after spontaneous or MOG-accelerated evolution of EAE [33,34,35,36,37,38,39,40,41]. The appearance of auto-Abs hydrolyzing DNA, proteins and oligosaccharides was revealed as the earliest and statistically significant and undoubtedly important marker of the beginning of many autoimmune diseases in humans and mice prone to AIDs (for review, see [5,6,7,8,9,35,36,37,38,39,40,41]). Enzymatic activities of abzymes are veraciously detectable before the appearance of typical known medical and biochemical markers of different AIDs at the pre-disease stage [5,6,7,8,9,35,36,37,38,39,40,41]. At the pre-disease stage and onset of different AIDs, the concentrations of different auto-Abs usually correspond to the indices spans, which are typical for healthy humans and experimental mice. The emergence of abzymes may authentically testify about the beginning of AIDs, while the increase in their enzymatic activities is coupled with the development of deep pathologies [5,6,7,8,9,35,36,37,38,39,40,41]. In this work, we analyzed the changes in the catalase activity of antibodies at the early stages of the development of EAE in C57BL/6 mice.

As shown earlier, during the spontaneous development of EAE, there is a relatively slow but gradual increase in hydrolysis efficiency by antibodies of DNA, MBP, MOG, and five histones [35,36,37,38,39,40]. Immunization of mice with MOG leads to a significant acceleration in EAE development and a sharp substantial increase in the activity of all abzymes. In this case, after mice immunization, three main stages of EAE development can be distinguished: the onset (6–8 days), the acute phase (18–20 days), and remission (>26–30 days). Already for 7 days, there is a significant increase in the activities of abzymes, which achieve maximum significance in the acute phase, and in the stage of remission, there may be a slight or moderate decrease in their activities [35,36,37,38,39,40]. Therefore, the analysis of miRNA hydrolysis by abzymes of EAE mice can provide additional opportunities for understanding at what stages of this disease development the accumulation of abzymes against miRNAs can occur.

Considering the meaningful role of miRNAs in the proliferation, differentiation, and maturation of different neuronal cells and the possible role of miRNAs in the development of MS, we study the miRNA-hydrolyzing activity of IgGs of EAE-prone C57BL/6 mice in time before and after their immunization with MOG. The substrate specificity of mice antibodies in the miRNA splitting was compared with that of MS patients.

## 2. Results

### 2.1. RNase Activity of IgGs

It was of interest whether, as for IgGs of patients with MS, SLE, and schizophrenia [48,49,50,51], antibodies from the blood of EAE-prone C57BL/6 mice can exhibit RNase activity. The EAE development in C57BL/6 mice occurs spontaneously and may be accelerated by immunizing mice with MOG [35,36,37,38,39,40,41]. In addition, it was important to understand at what stages of mice EAE development abzymes that hydrolyze some micro-RNAs may appear.

To study RNase activity of mice IgGs, we have used IgGs from the blood plasma of C57BL/6 mice corresponding spontaneous and MOG-induced EAE development. It is known that catalysis of various reactions by enzymes and abzymes occurs only after the formation of their specific complexes with substrates. First, it was shown that even IgGs of 3-months old mice possess RNase activity. The IgG_mix_ (14 µg; a mixture of 14 mice preparations) was subjected to SDS-PAGE under non-reducing conditions without DTT.

IgG_mix_ was electrophoretically homogeneous (Figure 1A). The relative RNase activity (RA, %) in the hydrolysis of miRNA was estimated using eluates of 2–3 mm gel fragments (Figure 1B). The positions of RNase activity correspond to gel fragments containing intact IgGs, with no other protein bands or peaks of catalytic activities. Canonical RNases have vastly lower molecular masses (13–15 kDa) than IgGs (150 kDa). Thus, the coincidence of the positions of RNase activity peak and protein band of IgGs directly indicates that mice IgG_mix_ hydrolyze miRNA, and it is not contaminated with classical RNases.

Previously, we have shown that IgGs of CBA and BALB mice not prone to the development of autoimmune diseases do not have catalytic activities in the hydrolysis of DNA and RNA [35,36,37,38,39,40,41,42,43,44]. IgG antibodies of these lines of mice, isolated using the method developed by us to obtain antibodies containing no impurities of canonical enzymes, were used as a control.

### 2.2. Hydrolysis of Homo-Oligonucleotides

To analyze the sites of RNAs hydrolysis, we used three fluorescently 5′-labeled homo-oligonucleotides (ONs) 5′-Flu-(pC)_23_, 5′-Flu-(pU)_23,_ and 5′-Flu-(pA)_23,_ as well as six 5′-Flu-miRNAs; two neuroregulatory miRNAs (miR-219a-5p, miR-9-5p) and four of immunoregulatory miRNAs (miR-21-3p, miR-146a-3p, miR-155-5p, and miR-326) characterized by impaired expression in patients with human multiple sclerosis [28].

Several preparations of IgGs from different 3-months-old mice (zero time; the beginning of the experiment) corresponded to the spontaneous development of EAE disease during 40 days after the beginning of the experiment and 7–37 days after their immunization with MOG. As an example, Figure 2 shows typical data of hydrolysis homo-oligonucleotides (ONs) by some of the 35 IgG preparations used and IgGs of CBA and BALB mice.

All IgGs hydrolyze 5′-Flu-(pC)_23_, 5′-Flu-(pU)_23,_ and 5′-Flu-(pA)_23_, almost non-specifically at nearly all their internucleoside bonds. 5′-Flu-(pC)_23_ was hydrolyzed, leading mainly to short ONs (Figure 2A). In the case of 5′-Flu-(pU)_23_, formation ONs of completely different lengths with comparable efficiency (Figure 2B). The maximum rate of hydrolysis was observed for (pC)_23_ and significantly lower for (pU)_23_; (pA)_23_ hydrolysis was very weak. To detect (pA)_23_ hydrolysis, a higher concentration of IgGs and longer incubation times were used (Figure 2C). Incubation of (pA)_23_ with antibodies for 7–10 h leads to the formation of hydrolysis products with the formation of patterns similar to those for(pU)_23_.

Under the conditions used, (pC)_23,_ which is the best substrate for IgGs of C57BL/6 mice was not hydrolyzed by antibodies from CBA and BALB mice not prone to spontaneous development of autoimmune diseases before and 20 days after their immunization with MOG (Figure 2A). IgGs from plasma of CBA and BALB mice do not have RNase activity.

### 2.3. Hydrolysis of Micro-RNAs

Unlike homo-oligonucleotides, antibody-dependent hydrolysis of six miRNAs was site-specific but by varying degrees. Taking into account the different rates of various miRNAs cleavage by different IgGs to determine the common sites of hydrolysis of each miRNA, IgGs were used at different concentrations. The identification of major, moderate, and weak hydrolysis sites was carried out based on averaged data for sites of miRNAs cleavage by all 35 IgG preparations of IgGs corresponding to different stages of EAE development.

The hydrolysis of miR-326 was exclusively site-specific with all 35 preparations (7 preparations corresponding to each time of blood sampling after immunization of mice with MOG: 0, 7, 12, 23, and 37 days). Figure 3A demonstrates typical patterns of miR-326 splitting by 13 of 35 preparations.

During the hydrolysis of this micro-RNA, only one major product (site of the hydrolysis G10-C11) and one minor (C18-A19) product were formed.

Some miRNAs’ hydrolysis efficiency could be very different for various IgG preparations. Figure 3B shows the data of miR-219a-5p hydrolysis by different IgGs at the same concentration. One can see that one of the preparations (lane 6) hydrolyzes this miRNA much more efficiently than other IgGs. Nevertheless, all 35 preparations showed five hydrolysis sites, two of which were major (C9-A19 and A10-A11), two minor (U5-G6 and G6-U7), and one, according to the averaged data for 35 preparations, can be attributed to the moderate site (C15-C15). As in the case of homooligonucleotides, IgG antibodies of CBA and BALB mice not prone to autoimmune diseases did not hydrolyze heterooligonucleotides Flu-miR-326 and Flu-miR-219-5p before and after mice immunization with MOG (Figure 3A,B).

Figure 4 shows sites of hydrolysis by IgGs of Flu-miR-21-3p and Flu-miR-155-5p micro-RNAs.

Six specific hydrolysis sites were found for hydrolysis of -21-3p with all antibody preparations (Figure 4A). However, in this case, four sites of specific hydrolysis were classified as major (A3-C4, A5, C6, C7-A8, and A8-G9) and two as moderate (G11-G12 and G9-G10).

Specific hydrolysis of miR-155-5p by all 35 IgGs proceeded at eight sites but with somewhat different efficiencies (Figure 4B). Five hydrolysis sites could be attributed to major (A4-U5, G6-C7, U8-A9, A9-A10, and U17-A18) and one to undoubtedly minor (U5-G6) sites. For most preparations, these sites should have been classified as minor, but for three of the 35 IgGs, they were major (for example, lanes 7 and 9; Figure 4B). Hydrolysis at two clearly detectable sites (U14-G15 and C12-G13) was very different for IgGs preparations from the blood of other mice.

Hydrolysis of miR-146-3p with all IgG preparations was site-specific at four major (U11-C12, C12-A13, C17-U18, and C20-A21) and one very minor site (C2-U3) (Figure 5A).

A completely different picture was found in the case of miR-9-5p (Figure 5B). All antibodies hydrolyzed this miRNA with somewhat different efficiency. Based on averaging data for 35 preparations, three sites can be attributed to conditionally major (U8-U9, U9-A10, and C12-U13) and three minor sites (G6-G7, G18-U19, and A20-U21). However, in parallel with specific hydrolysis of this miRNA at six sites, nonspecific hydrolysis at all internucleoside phosphate groups of miRNA (Figure 5B), as in the case of homo-oligonucleotides (Figure 2), was observed.

### 2.4. Spatial Structures of miRNAs

The spatial structures of six micro-RNAs having minimal free energy were calculated earlier [48,49,50,51]. The relative percent of every product of every micro-RNA splitting by individual IgGs was calculated. Then, using the data of three independent experiments for each IgG sample, the average percentage of each product corresponding to seven blood plasma IgGs was calculated. Figure 6 and Figure 7 demonstrate the location of splitting sites in the spatial structures of six micro-RNAs in the case of IgG antibodies.

As mentioned above (Figure 2, Figure 3, Figure 4 and Figure 5), IgGs from plasmas of various mice hydrolyze six micro-RNAs with different efficiencies and, sometimes, at different sites. Finally, Figure 6 and Figure 7 demonstrate averaged data on the efficiency of six micro-RNAs hydrolysis by seven IgGs at each of the sites.

The main sites for more efficient cleavage of miR9-5p by IgGs are located in the specific loop of this micro-RNA and its 3′-terminal part (Figure 6A). The major hydrolysis site of miR-148a-3p is also located in the specific loop (16.5%), but hydrolysis of this miRNA mainly occurs in the duplex structure and its 5′-terminal region (Figure 6B). One major miR-219a-5phydrolysis site is also located in a specific loop, and six of the eight others are distributed over the duplex part of the molecule, its 3′ and 5′ terminal parts (Figure 6C). The most significant number of hydrolysis sites was found for miR-326 (Figure 6D). Interestingly, two major sites (10.7 and 7.5% of the hydrolysis) and six other sites correspond to 5′ and 3′ duplex zones of this micro-RNA and only sites to its specific loop.

In the case of miR-21-3p, hydrolysis sites in the specific loop and duplex part are absent and are mainly located in its 5′-terminal part of this RNA (Figure 7A).

Nearly the same situation is observed for the hydrolysis of miR155-5p (Figure 7B); the main sites of hydrolysis are localized in the 5′-terminal part of this micro-RNA. Thus, the immune response (and antibody production) against each of the microRNAs is highly specific and individual.

### 2.5. Affinity of IgGs for Micro-RNAs

To evaluate the affinity of IgGs to micro-RNA, miR-155-5p and the conditions corresponding to the pseudo-first-order reaction were used-linear sections of the rate from the concentration of IgGs (0.2 μM) and the reaction time. Evaluation of the values of *K*_m_ and *V*_max_ (*k*_cat_) was performed using miR-155-5p and three IgG preparations corresponding to 7 days after the mice immunization with MOG (Figure 8). In the case of all three preparations, the same *K*_m_ values were obtained, 5.8 ± 1.7 µM. At the same time, these three IgG preparations showed different *k*_cat_ (*k*_cat_ = *V*_max_ (M/min)/[IgGs], M) values: 0.06 ± 0.01, 0.19 ± 0.03, and 0.25 ± 0.05 min^−1^ (Figure 8).

### 2.6. In Time Changes of Micro-RNAs Hydrolysis during the Development of EAE 

The EAE development in C57BL/6 mice occurs spontaneously and may be accelerated by immunizing mice with MOG [35,36,37,38,39,40]. There are several stages of EAE development after mice immunization with MOG or complex DNA with histones: the onset at 7–8, the acute phase at 18–20, and the remission stage ≥25–30 days. The spontaneous and accelerated EAE development occurs as the result of certain changes in bone marrow HSCs differentiation profiles and an increase of lymphocyte proliferation in different organs associated with parallel production of abzymes splitting DNAs, MBP, MOG, and histones [35,36,37,38,39,40]. Appendix A show the changes in the differentiation profile of the bone marrow stem cells and lymphocyte proliferation in different organs of C57BL/6 mice during the spontaneous, MOG- and DNA-histones complex-induced development of EAE.

The blood of mice was collected at various times up to 40 days after the start of the experiments (time zero) before and after mice immunization; days of blood sampling are shown in the Figures.

The relative activity of abzymes hydrolyzing DNA, MOG, MBP, and histones during the spontaneous development of EAE in C57BL/6 mice increases slowly and smoothly—almost linearly [35,36,37,38,39,40]. A strong increase in the relative activities (RAs) of abzymes hydrolyzing these antigens-substrates occurs as early as 7 days after immunizing of mice with MOG and reaches their maximum values during the acute phase (18–20 days). During the period of remission, the relative activities of abzymes may decrease slightly or moderately [35,36,37,38,39,40,41]. Appendix A demonstrates a very strong increase in the RAs of ABZs in the hydrolysis of DNA, MOG, and myelin basic protein beginning from 7 days and the formation of antibodies with maximum activity by 20 days after the immunization of mice with MOG.

It was interesting to analyze the change in the time of development of EAE in the relative activity of antibodies in the hydrolysis of micro-RNAs compared with the hydrolysis of DNA, MBP, and MOG before and after mice immunization with MOG. With this in mind, IgG antibodies were obtained from six groups of three-month-old C57BL/6 mice (seven mice per group), corresponding to 40 days of spontaneous development of EAE.

Figure 9 shows data on changes in the average activity of abzymes corresponding to seven different IgGs of each group of mice in the hydrolysis of nine RNA-substrates before and after immunization of mice with MOG. In the case of spontaneous development of EAE, an absolutely unexpected result was obtained. While the activities of antibodies in the hydrolysis of DNA, MBP, and MOG in the process of spontaneous development of EAE increased intermittently but constantly (Appendix A), starting from three months of mice age, the relative activity in hydrolysis of all nine RNA-substrates decreased slowly (Figure 9).

At 3 months of life, the average relative activity of antibodies very strongly depended on the RNA-substrate and decreased in the following order: (pC)_23_ > miR-21-3p > miR-155-5p > miR-219a-5p ≈ miR-146a-3p > miR-9-3p > miR-326 ≈ (pU)_23_ > (pA)_23_ (Figure 9). Over 20 days after the start of the experiment, due to the spontaneous development of EAE, the decrease in the relative activity also depended on the RNA substrate (approximate % of initial values): (pC)_23_ (~98–99) > (pA)_23_ (75) > miR-219a-5p (57) ≈ miR-326 (55) > miR-155-5p (42) > miR-21-3p (20) ≈ miR9-5p (18) > (pU)_23_ (5) > miR-219a-5p (≈0).

As showed in previous studies, the production of antibodies with various enzymatic activities is the earliest and statistically significant indicator of the development of autoimmune reactions (for review, see [4,5,6,7,8,9,10]. In the case of the initial onset stage of development of different AIDs, abzymes are detected at a time when the titers of autoantibodies to various self-antigens still correspond to the ranges of their variations in conditionally healthy donors. The detection of abzymes that hydrolyze DNA, MBP, MOG, and histones in mice as early as three months of life indicated that already at this age, the immune status of C57BL/6 mice is violated, and they demonstrated the initial forms of EAE. Previously, we used only 3-month-old C57BL/6 mice to study the mechanisms of EAE development. However, in this work, given the opposite nature of the change—a decrease in the relative activity of antibodies in micro-RNA hydrolysis during the spontaneous development of EAE in comparison with an increase in the hydrolysis of DNA, MBP, and MOG [35,36,37,38,39,40,41], we obtained an additional special set of IgGs from mice corresponding 50, 80 and 92 days (3 months) after their birth.

The change in relative activity from 50 to 92 days of life in mice was highly dependent on RNA analyzed. The most significant decrease in antibody activity (79%) during this time was observed in hydrolysis (pC)_23_ (Figure 10A).

Moreover, a strong statistically significant decrease in activity from 50 days to 3 months of mice life occurred in the case of (%): (pA)_23_ (81), (pU)_23_ (66), miR-9-5p (41) (Figure 10B), miR-146a-3p (62) (Figure 10C). A slight decrease in the relative activity of IgGs occurred in the case of miR-21-3p (26%; Figure 10A). There was no significant change in activity from 50 to 92 days in the case of miR-326 (Figure 10B), while a noticeable change in the hydrolysis of miR-219a-5p was observed only from 80 to 92 days (Figure 10C). Only in the case of miR-155-5p, during this period, there was an increase in the activity of antibodies in the hydrolysis by 25% (Figure 10C).

Thus, in general, there is a tendency to reduce the relative activity of most of the abzymes that hydrolyze micro-RNAs in the period from 50 to 132 days of mice life. Therefore, it was interesting to compare the effect of MOG treatment of mice on RAs of IgGs in the hydrolysis of DNA and proteins with activity in the splitting of RNAs. Figure 9 shows data on changes in the activity of IgGs in the hydrolysis of nine RNAs before and after immunization. The spontaneous development of EAE leads to a significant decrease in the activity of antibodies in the hydrolysis of six out of nine RNAs. However, at the same time, in the case of two RNAs (miR-219a-5p and (pU)_23_), a slight temporary increase in antibody activity is observed at 10 days after the start of the experiment. After a noticeable decrease in the efficiency of miR-9-5p hydrolysis by 10 days (Figure 9C), a constant increase in the activity of IgGs is observed. Nevertheless, despite these features in the case of three RNAs, immunization of mice in all cases leads to a statistically significant (*p* < 0.05) sharp increase in the activity of antibodies in the hydrolysis of all nine RNAs compared to their activity at 3 months of age (-fold): (pA)_23_ (5.2), (pU)_23_ (4.6), miR-219a-5p (3.9), miR-9-5p (3.1), miR-326 (2.9), miR-146a-3p (2.5), miR-155-5p (1.9), miR-21-3p (1.2), and (pC)_23_ (1.2).

As mentioned above, immunization of mice with MOG leads to a significant increase in the activity of antibodies in the hydrolysis of DNA, MOG, MBP, and histones. It is very important that the maximum activity in the hydrolysis of these four immunogens-substrates is observed in the acute phase of the disease—20 days (Appendix A). A specific singularity of the sharp increase in the activity of IgGs in the splitting of RNAs and micro-RNAs after mice immunization with MOG is that the maximum of their activity is observed mainly in 7 and in some cases in 7–14 days—the initial stage of the pathology development (Figure 9). Then, by the 20th day of the acute phase of EAE, there is a strong decrease in the RAs of abzymes in the hydrolysis of all nine RNAs. Thus, immunization of C57BL/6 mice with MOG stimulates not only the production of abzymes that hydrolyze DNA, MOG, MBP, and histones but also homo-RNAs and micro-RNAs.

## 3. Discussion

As shown earlier, in SLE and EAE, in comparison with the norm before the disease, the first change in the differentiation profile of bone marrow stem cells occurs in the first stage of spontaneous or specific antigen-induced pre-disease conditions, and then, when moving to a deep pathology, an additional change of differentiation profile occurs (for review see [4,5,6,7,8,9,10]). These changes are associated with the appearance in the blood of mice of abzymes that hydrolyze DNA, MBP, ATP, and polysaccharides. It should be noted that the relative activity of antibodies from the cerebrospinal fluid of multiple sclerosis patients in the hydrolysis of DNA, MBP, and polysaccharides, depending on the substrate, is 30–60 times higher than that from the blood of the same patients [22,23,24]. Taking this into account, we believe that the development of autoimmune diseases can begin at the level of the cerebrospinal fluid and the brain.

Previously, it was shown that antibodies from healthy donors are inactive in the hydrolysis of RNA and DNA [4,5,6,7,8,9,10]. At the same time, IgGs with DNase, RNase, proteolytic, and amylase activity are the earliest statistically significant markers of several autoimmune pathologies [4,5,6,7,8,9,10].

This work first showed that the hydrolysis of homo-RNAs and micro-RNAs, as in the case of patients with various AIDs [4,5,6,7,8,9,10], is an intrinsic property of IgGs of C57BL/6 mice predisposed to EAE. At the same time, hydrolysis of three homo-RNAs occurs non-specifically, while six micro-RNAs proceed at specific sites.

A feature of miRNAs hydrolysis is that the activity of Abs is detected as early as 50 days after the birth of mice. In the period from 50 to 92 days (3-month-old mice), there is a significant decrease in the activity of IgGs in the hydrolysis of 8 out of 9 RNAs. Within 40 days after the start of the experiment, a further reduction in the activity of ABZs in the hydrolysis of miRNAs is mainly observed. This result is entirely inconsistent with the slow, gradual increase in the activity of antibodies in the hydrolysis of DNA, MBP, MOG, histones, and DNA during the spontaneous development of EAE in mice (Appendix A) [35,36,37,38,39,40,41].

The constant production of abzymes that hydrolyze DNA, MBP, and histones during the spontaneous development of EAE is associated with some features of these autoantigens. The main antigen for producing antibodies against DNA and histones are DNA complexes with histones, which appear in the blood as a result of cell apoptosis [45]. Anti-DNA abzymes easily penetrate the outer and nuclear membranes of cells, hydrolyze the DNA of nuclear chromatin, and stimulate cell death through apoptosis [46,47]. This leads to an increase in the concentration of histones and DNA in the blood and antibodies-abzymes against them and, as a result, to the acceleration of several AIDs development [4,5,6,7,8,9,10]. The peculiarity of the production of abzymes that hydrolyze MBP is that histones and MBP have a high level of homology of their protein sequences. This leads to the fact that abzymes against histones effectively hydrolyze MBP and against MBP, on the contrary, all five histones ([52,53,54] and refs. therein). Since histones constantly appear in the blood due to cell apoptosis, this leads to a violation of the immune system and the synthesis by lymphocytes of Abs-abzymes against MBP. These factors underlie the continuous synthesis by lymphocytes producing antibodies against DNA, MBP, and histones and abzymes hydrolyzing these autoantigens [8,9,10].

One cannot exclude that the very important difference between abzymes against DNA (MBP and histones) and miRNAs is that micro-RNAs have many different biological functions, including regulating up to several hundred genes [55,56]. Different changes in microRNAs (microRNA-regulated gene networks) could result in the realignment in the expression of many genes in different cells. At various stages, the mice’s growth processes should be regulated by different micro-RNAs. It has been shown that the expression of many miRNAs changes with age. For example, CD1 mice show decreased expression of miR-148, miR-219, miR-199a, miR-214, miR-335, miR-411, and other micro-RNAs in the lungs of postweaning females and adult females compared to neonatal mice [57]. Similar data on decreased micro-RNA expression with age were obtained in peripheral blood mononuclear cells and plasma in humans [58,59]. However, some miRNAs’ expression may increase with age [59].

At the initial stages of the mice’s growth, the concentration of some miRNAs may be relatively high but then gradually decreases with mice age. A decrease in the concentration of miRNAs with age should lead to a reduction in the concentration of abzymes against these miRNAs. Therefore, in 50-day-old mice, the increased activity of abzymes in the hydrolysis of micro-RNAs may be associated with the production of antibodies-abzymes against micro-RNAs in mice in increased concentrations.

Immunization of mice with MOG leads to severe impairment of their immune system leading to specific violations of an extended nature. A change in the profile of bone marrow stem cells differentiation leads to the production of lymphocytes that produce abzymes hydrolyzing not only MOG but also in parallel DNA, myelin basic protein, and histones. It cannot be ruled out that such changes in the differentiation profile may, in parallel, also lead to the production of lymphocytes producing abzymes to other external and self-antigens, including micro-RNAs. In this case, at the initial stage of EAE accelerated development after mice immunization with MOG, there may be the appearance of abzymes with higher activity in micro-RNAs hydrolysis. However, the effect of MOG on increased production of lymphocytes that synthesize anti-micro-RNAs antibodies with catalytic activity may be temporary. In connection with this, it should be noted that, in 7–14 days after mice immunization with MOG, the activity of abzymes hydrolyzing nine RNAs increases only 1.2–5.2-fold (Figure 9). At the same time, by the 20th day after immunization, the DNase activity of IgGs increases by 25 times [35,36,37,38]. It seems that the general trend of decreasing activity of some abzymes hydrolyzing miRNAs with age of mice leads to the significant reducing of the rise in their activity due to mice immunization with MOG.

The general trend towards a decrease in the activity of abzymes hydrolyzing micro-RNAs over time due to a reduction in the concentration of micro-RNAs with age can probably lead to a recession in such antibodies as early as 20 days after immunization, which is observed in the experiment.

## 4. Materials and Methods

### 4.1. Materials and Chemicals

All preparations were free from possible contaminants. All high-quality chemicals were from Sigma (St. Louis, MO, USA). Sorbents columns (Superdex 200 HR 10/30 (17-5175-01)) and Protein G-Sepharose (17061801)) were purchased from GE Healthcare (GE Healthcare, New York, NY, USA).

### 4.2. Experimental Animals

We recently used inbred 3-months-age C57BL/6 mice to study possible mechanisms of spontaneous and MOG-induced EAE development [35,36,37,38,39,40,41]. The blood of mice was collected at various times up to 40 days after the start of the experiments (time zero) before and after mice immunization (0, 7, 12, 14, 20–23, and 37 days); days of blood sampling are shown in the Figures. All groups used consisted of 7 mice.

Here, we also obtained an additional group of mice and their antibodies corresponding to 50, 80, and 92 days-old mice. These mice were grown in a special mouse vivarium of the Institute of Cytology and Genetics (ICG) using special conditions free of any pathogens. We also used seven-month-old CBA (CBAxC57BL-F1) and BALB mice not prone to spontaneous development of autoimmune diseases.

All experiments with C57BL/6, CBA and BALB mice were carried out according to the protocol of the ICG Bioethical Committee (document number 134A of 7 September 2010), fulfilling the humane principles for operating with animals established by the Directive of European Communities Council (86/609/CEE). The Bioethical Committee of the institute has supported this study.

### 4.3. Antibody Purification

Electrophoretically homogeneous polyclonal IgG antibodies from the plasma of mouse blood were first purified using affinity chromatography proteins of blood plasmas (7 mice in each group) on Protein G-Sepharose. In addition, they were isolated using FPLC gel-filtration (Fast protein liquid chromatography-gel filtration) on Superdex 200 HR 10/30 column [35,36,37,38,39,40,41]. After gel filtration, central parts of IgG preparations peaks were subjected to filtration through special filters (pore size 0.1 µm). In addition, 7 preparations of homogeneous IgG preparations were obtained from the blood plasma of CBA and BALB mice not prone to autoimmune diseases.

The IgG preparations were subjected to the assay RNase activity after their SDS-PAGE using all eluates of gel fragments as in [35,36,37,38,39,40,41] to exclude possible traces of canonical RNases. It was shown that only intact IgG-antibodies demonstrate RNase activity, and no other protein bands or ribonuclease activities in different fragments of gel were found. The Appendix A give a more detailed description of these experiments.

### 4.4. SDS-PAGE Assay of RNase Activity

SDS-PAGE analysis of Abs for homogeneity was performed using a 5–16% gradient gel containing 0.1% sodium dodecyl sulfate (SDS; Laemmli system) as in [35,36,37,38,39,40]. The IgGs were visualized by silver staining.

Analysis of RNase activity of IgGs after SDS-PAGE was performed as in [48,49,50,51]. IgGs (10–40 μg) were pre-incubated at 30 °C for 35 min under non-reducing (1% SDS, 50 mM Tris-HCl, pH 7.5, and 10% glycerol) conditions. After SDS-PAGE electrophoresis of Abs to restore the RNase activity of IgGs, SDS was removed by incubating the gel for 1 h at 30 °C with 4.0 M urea and washed 12 times (8–10 min) with H_2_O. Then 2.5–3-mm cross sections of the gel long slices were cut up and then incubated with 50 μL 52 mM Tris-HCl (pH 7.5) supplemented with 50 mM NaCl for 7 days at 4 °C to allow IgGs refolding and eluting them from the gel. The solutions were removed from the gels using centrifugation and used for the assay of RNA hydrolysis, as described in the article. Parallel control lanes of gel were used for the detection of the position of IgG on the gel by silver staining.

### 4.5. Analysis of Homo-Oligonucleotides and miRNAs Splitting by IgGs

Fluorescein isothiocyanate (Flu) fluorescently labeled ribooligonucleotides 5′-Flu-(pA)_23_, 5′-Flu-(pU)_23_, and 5′-Flu-(pC)_23_, and several micro-RNAs characterized participating in the neuroinflammation regulation of by impaired expression in MS [28], were used in the study. Two neuroregulatory miRNAs are: miR-219a-5p (5′-Flu-UGAUUGUCCAAACGCAAUUCU) and miR-9-5p (5′-Flu-UCUUUGGUUAUCUAGCUGUAUGA). In addition, four immunoregulatory miRNAs were also used: miR-21-3p (5′-Flu-CAACACCAGUCGAUGGGCUGU), miR-146a-3p (5′-Flu-CCUCUGAAAUUCAGUUCUUCAG), miR-155-5p (5′-Flu-UUAAUGCUAAUCGUGAUAGGGGU), and miR-326 (5′-Flu-CCUCUGGGCCCUUCCUCCAG).

The reaction mixture (10–15 μL) contained 50 mM Tris-HCl pH 7.5, 0.05 mg/mL one of miRNAs (1.3–1.6 μM depending on the miRNA used), 0.005 mg/mL (33.3 nM) IgGs. All mixtures were incubated for 2 h at 37 °C. The reaction was brought to a stop by the addition of a buffer (10–15 μL) containing 8.0 M urea and 0.025% xylene cyanol, and the product of hydrolysis analyzed by 20% PAGE using denaturing conditions (8 M urea, 0.1 Tris, 0.1 M boric acid, and 0.02 M Na_2_EDTA; pH 8.3). The gels were explored using FLA 9500 Typhoon laser scanner (GE Healthcare, New York, NY, USA). The markers of RNAs molecular weights were obtained by statistical alkaline hydrolysis of 3.2 μM miRNAs (at all internucleoside bonds) by substrates incubation in bicarbonate buffer (50 mM, pH 9.5) for 15 min at 95 °C. The relative RNase activity was estimated from a decrease in intact RNA substrates. Mean values of the relative activities in the hydrolysis of all substrates by antibodies were calculated by averaging activity values corresponding to seven mice in each group of mice.

All initial rates of RNAs hydrolysis were measured using the reaction conditions of the pseudo-first-order corresponding to linear parts of dependencies on time (30–40% of RNAs hydrolysis) and concentrations of IgGs.

### 4.6. Spatial Models of microRNAs

The spatial models of six micro-RNAs (miR9-5p, miR219a-5p, miR-326, miR-155-5p, miR-21-3p, and miR-146a-3p) were generated previously [48,49,50,51] using Predict a Secondary Structure server: http://rna.urmc.rochester.edu/RNAstructureWeb/Servers/Predict1/Predict1.html (accessed on 14 February 2018), which uses a combination of four algorithms for predicting the secondary structure of RNA with minimal energy.

### 4.7. Statistical Analysis

The results corresponding to the average values (mean ± standard deviation) from three independent experiments for each preparation of IgGs and RNA-substrate, averaged over 7 different mice in every group.

Based on the data on the hydrolysis of each of the homooligonucleotides and micro-RNAs, an estimation of relative activity was made using the loss of each of the initial substrates after their incubation with each of the individual antibody preparations (some examples of substrates hydrolysis are shown in Figure 2, Figure 3, Figure 4 and Figure 5). Then, the average value of antibody activity (mean ± standard deviation) was calculated using relative activities of seven IgG preparations of each of the analyzed groups of mice and shown in the Figure 9 and Figure 10.

The efficiency of hydrolysis of all miRNAs at different sites by each of IgG preparations was estimated as a percentage relative to the sum of the relative efficiencies of spots (100%) of each of the hydrolysis products. The assessment of the relative average efficiency of hydrolysis in each of the sites was carried out by averaging the activity values for seven preparations of each of mice studied groups (Figure 6 and Figure 7).

The apparent *K*_m_ and *V*_max_ (*k*_cat_) values for RNA were calculated from the dependencies of *V* versus [RNA] by non-linear fitting using Microcal Origin v5.0 software. Errors in the values were within 7–15%. The differences between characteristics of IgG samples of various groups were estimated by the non-parametric Kruskal–Wallis one-way analysis of variance, *p* < 0.05 was considered statistically significant.

## 5. Conclusions

Here, we have first shown that IgG-abzymes from EAE-prone C57BL/6 mice efficiently hydrolyze nonspecifically at all sites homo-oligonucleotides and six miRNAs (miR-9-5p, miR-219a-5p, miR-326, miR-155-5p, miR-21-3p, and miR-146a-3p) in specific sites. During the spontaneous development of EAE from 50 to 132 days after birth, the relative activity of abzymes in the hydrolysis of all nine RNAs substrates is noticeably or significantly reduced. Immunization of three-month-old mice (at 92 days of age) with MOG leads to a sharp temporary increase in the activity of antibodies in micro-RNAs hydrolysis at 7 days (the initial stage of the disease), followed by a sharp decrease in their activity by 20 days. It was suggested that a temporary increase in the activity of abzymes in the splitting of micro-RNAs is associated with a tendency for a constant decrease in the concentration of micro-RNAs during mice growth and aging.

## Figures and Tables

**Figure 1 ijms-24-04433-f001:**
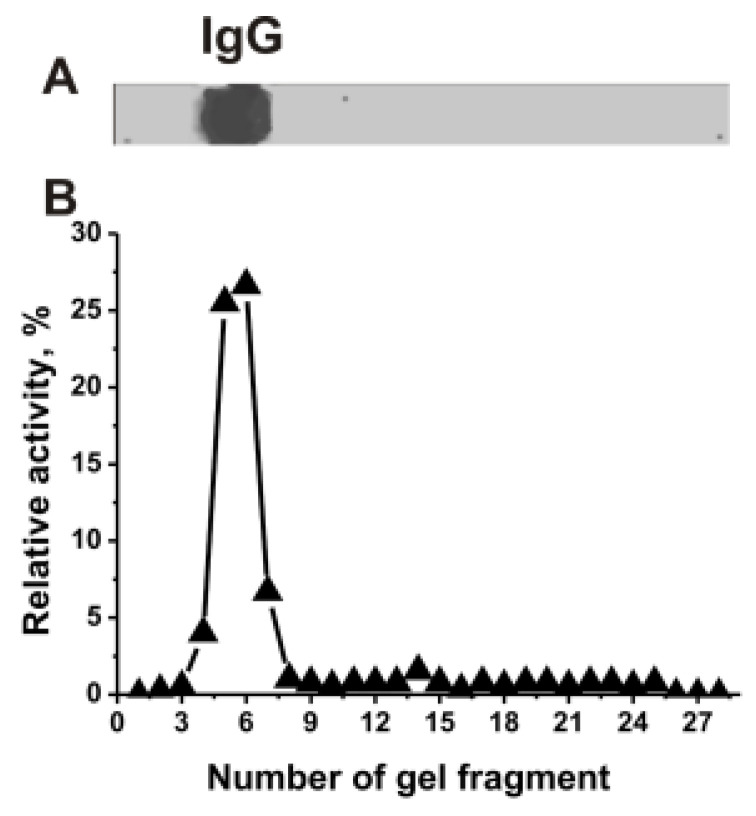
SDS-PAGE analysis of the electrophoretic homogeneity IgG_mix_ corresponding to a mixture of 15 individual antibodies in 4–18% gradient gel followed by silver staining (15 μg IgGs were used) (**A**). After SDS-PAGE electrophoresis similar to (**A**), the gel was incubated under special conditions for the renaturation of IgGs. The relative miR-326-hydrolyzing activity (%) was revealed using the extracts of 2.5–3-mm fragments of one longitudinal slice of the gel (**B**). The RA of IgG_mix_ corresponding to complete hydrolysis of miR-326 after 7 h of incubation with 5 µL of the eluate was taken for 100% (**B**). The error from two experiments in the initial rate determination did not exceed 7–10%.

**Figure 2 ijms-24-04433-f002:**
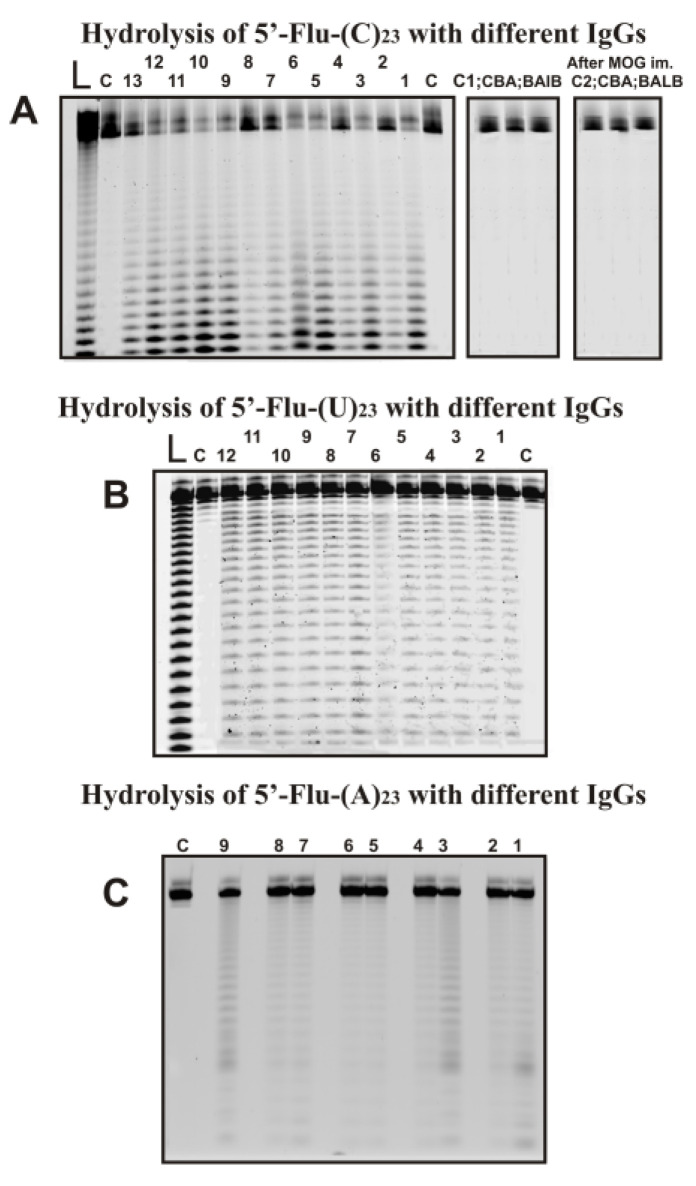
The patterns of Flu-(C)_23_ (**A**), Flu-(U)_23_ (**B**), and Flu-(A)_23_ hydrolysis by different IgG preparations (0.2–0.4 µM IgGs; 2 h of incubation) from plasmas of 9–13 different individual C57BL/6 mice and Flu-(C)_23_ (**A**) hydrolysis with IgGs of CBA and BALB mice before and 20 days after their immunization with MOG. Hydrolysis products were revealed due to the fluorescent residue (Flu) on 5′-ends of the homo-oligonucleotides. The lengths of the hydrolysis products are indicated on panels A-C. Lanes L correspond to ribooligonucleotides length markers, while all lanes C to ONs incubated without Abs (**A**–**C**).

**Figure 3 ijms-24-04433-f003:**
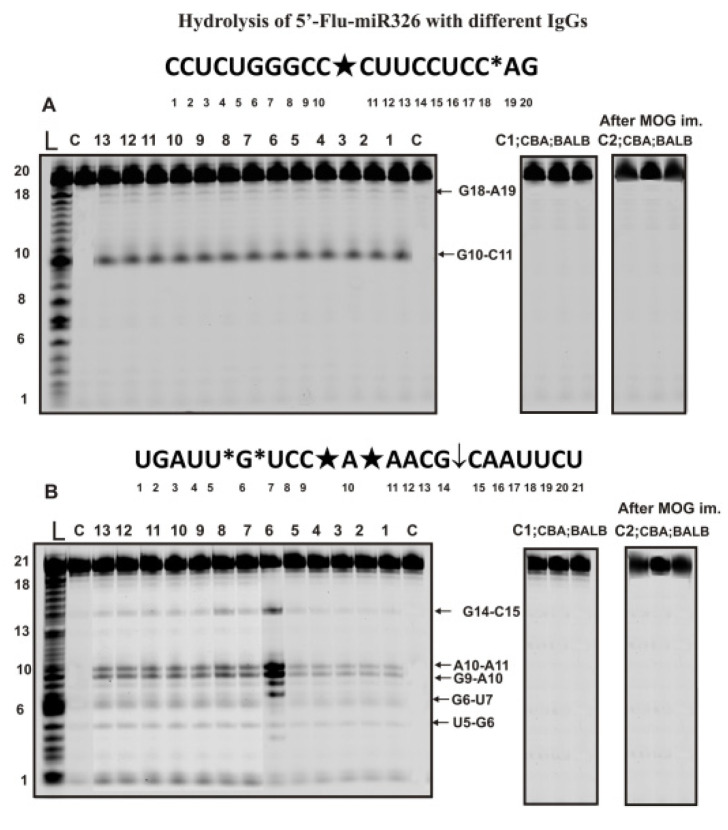
The patterns of Flu-miR-326 (**A**) and Flu-miR-219-5p (**B**) hydrolysis by 13 different IgG preparations (0.2 µM IgGs; 2 h of incubation) from the blood plasma of 13 different C57BL/6 mice corresponding to different phases of EAE development and these ONs hydrolysis with IgGs of healthy CBA and BALB mice before and 20 days after their immunization with MOG (**A**,**B**). Hydrolysis products were revealed due to the fluorescent residue (Flu) on 5′-ends of micro-RNAs. Lanes L correspond to control ONs of length markers, while lanes C, C1, and C2 to micro-RNAs incubated without Abs. The lengths of the hydrolysis products and sites of the hydrolysis are indicated on Panels (**A**,**B**). Panels (**A**,**B**) show miRNA sequences and their hydrolysis sites. Major hydrolysis sites are characterized by large stars (★), moderate by arrows (↓), and minor hydrolysis sites by small stars (*).

**Figure 4 ijms-24-04433-f004:**
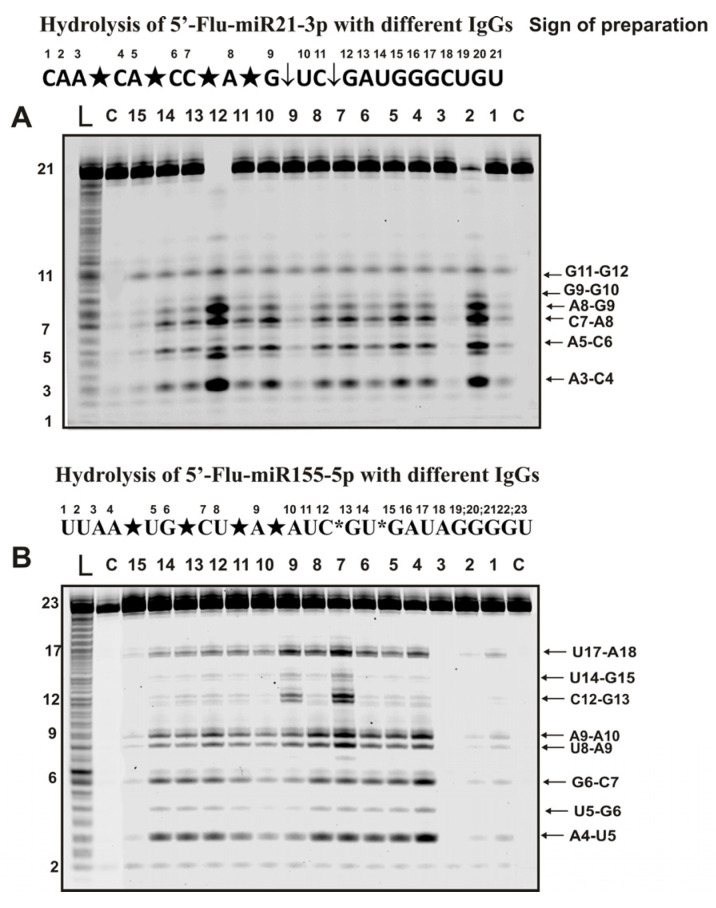
Hydrolysis products were revealed due to the fluorescent residue (Flu) on 5′-ends of micro-RNAs. The patterns of Flu-miR-21-3p (**A**) and Flu-miR-155-5p (**B**) hydrolysis by 13 different IgG preparations (0.2 µM IgGs; 2 h of incubation) from the blood plasma of 13 different mice corresponding to different phases of EAE development. Lanes L correspond to control ONs (length markers), while lanes C to micro-RNAs incubated without Abs. The lengths of the hydrolysis products and sites of the hydrolysis are indicated on Panels (**A**,**B**). Panels (**A**,**B**) show miRNA sequences and their hydrolysis sites. Major hydrolysis sites are indicated by large stars (★), moderate by arrows (↓), and minor hydrolysis sites by small stars (*).

**Figure 5 ijms-24-04433-f005:**
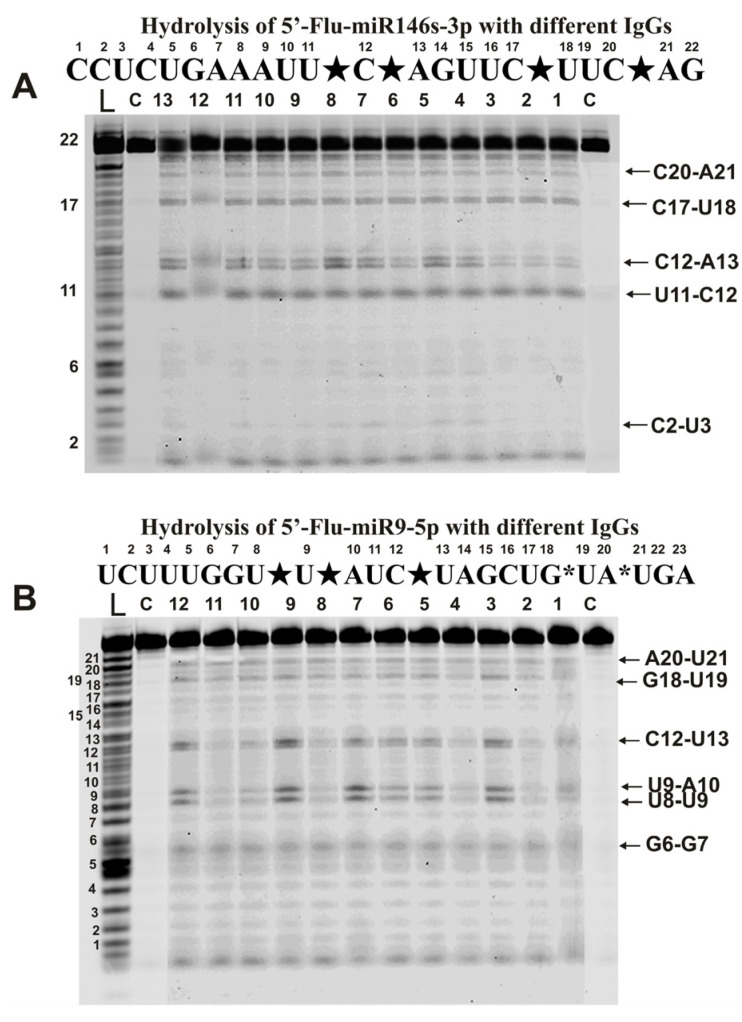
The patterns of Flu-miR-146s-3p (**A**) and Flu-miR-9-5p (**B**) hydrolysis by different IgG preparations (0.2 µM IgGs; 2 h of incubation) from the blood plasma of 12–13 different mice corresponding to different phases of EAE development. Hydrolysis products were revealed due to the fluorescent residue (Flu) on 5′-ends of micro-RNAs. Lanes L correspond to control ONs (length markers), while lanes C to micro-RNAs incubated without Abs. The lengths of the hydrolysis products and sites of the hydrolysis are indicated on Panels (**A**,**B**). Panels (**A**,**B**) show miRNA sequences and their hydrolysis sites. Major hydrolysis sites are indicated by large stars (★), and minor hydrolysis sites by small stars (*).

**Figure 6 ijms-24-04433-f006:**
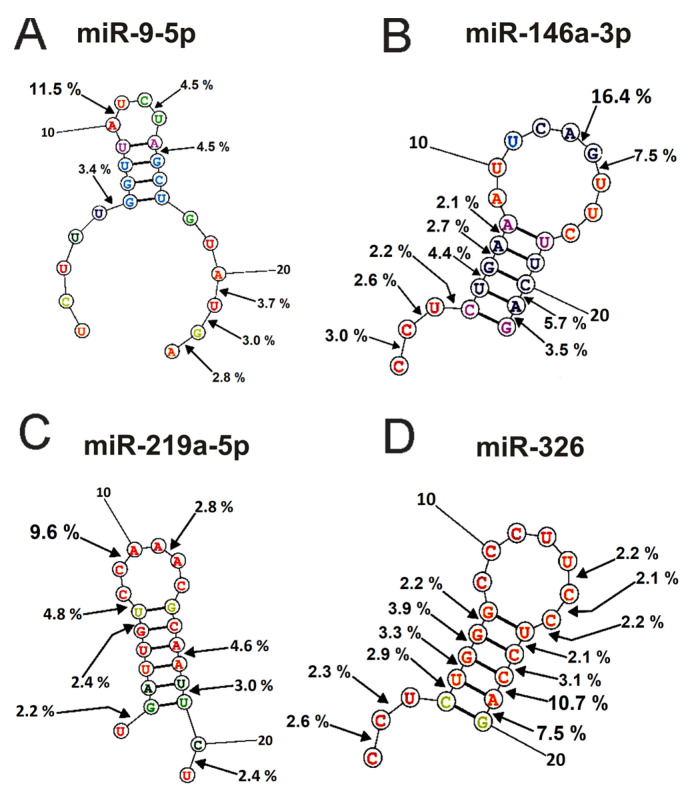
The average efficiency of Flu-miR-9-5p (**A**), Flu-miR-146a-3p (**B**), Flu-miR-219a-5p (**C**), and Flu-miR-326 (**D**) hydrolysis by IgGs from mice blood plasmas in different sites of their cleavage. The average percent of micro-RNAs hydrolysis and the position of different splitting sites in micro-RNAs are shown using spatial models of four micro-RNAs.

**Figure 7 ijms-24-04433-f007:**
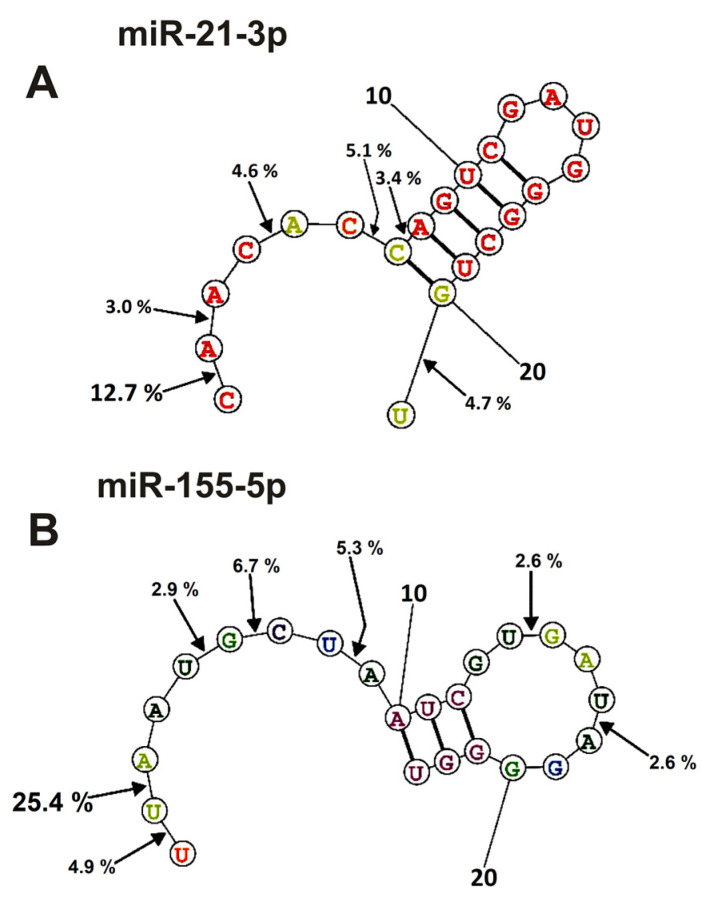
The average efficiency of Flu-miR-21-3p (**A**) and Flu-miR-155-5p (**B**) hydrolysis by IgGs from mice blood plasmas in different sites of their cleavage. The average percent of micro-RNAs hydrolysis and the position of different sites in micro-RNAs are shown using spatial models of four micro-RNAs.

**Figure 8 ijms-24-04433-f008:**
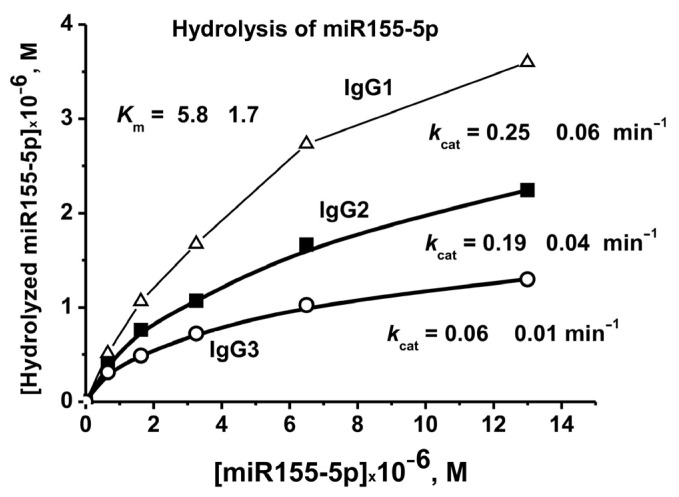
The evaluation of apparent *K*_m_ and *V*_max_ (*k*_cat_) values for miR-155-5p using dependencies of *V* versus [RNA] by non-linear fitting using Microcal Origin v5.0 software.

**Figure 9 ijms-24-04433-f009:**
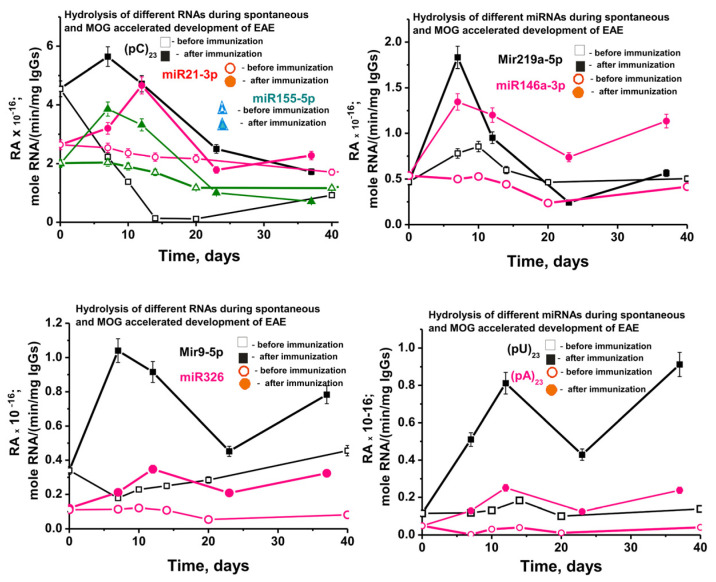
In time changes of an average relative activity of IgGs (seven mice in each group) in the hydrolysis of four micro-RNAs and three homo-ONs before (spontaneous development of EAE) and after 3-month-old mice immunization with MOG. All RNA-substrates are indicated on the Panels of the Figure.

**Figure 10 ijms-24-04433-f010:**
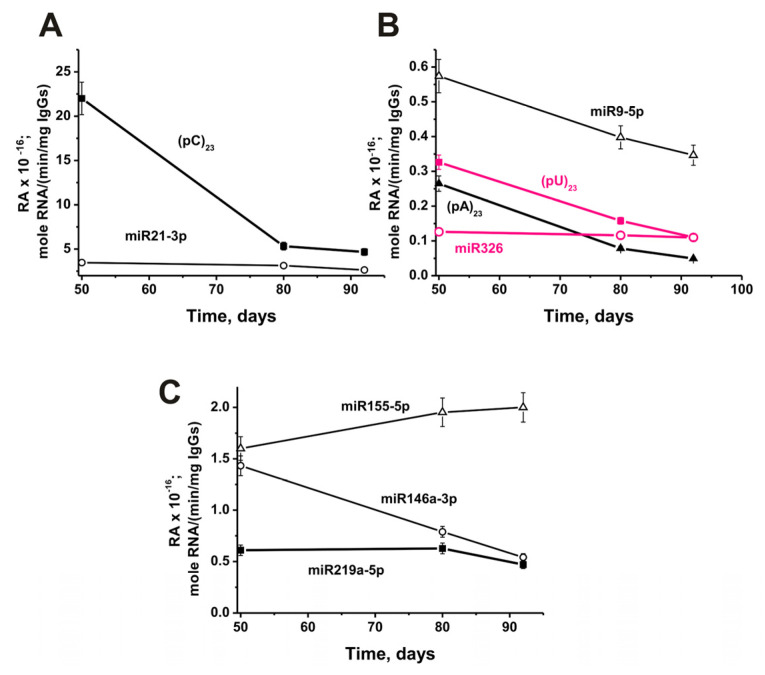
In time changes of an average relative activity of IgGs (seven mice in each group) in the hydrolysis of four micro-RNAs and three homo-ONs using 50 to 92 days (3-month-old) mice after their birth (before beginning of experiments with 3-month-old mice). All RNA-substrates are indicated on the Panels of the Figure (**A**–**C**).

## Data Availability

The data that supports the results are included in the article and its Appendix A.

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
