# Peer review of "Experimental Autoimmune Encephalomyelitis of Mice: IgGs from the Sera of Mice Hydrolyze miRNAs"

_ijms, 2023, doi:10.3390/ijms24054433_

Round 1
Reviewer 1 Report (Previous Reviewer 2)
Thank you for point-by-point responding my comments. The revised manuscript is a great improvement. I think this manuscript will be acceptable.
Author Response
There are no new remarks
Soncerely
Georgy Nevinsky
Reviewer 2 Report (Previous Reviewer 1)
The issues with the present report:
1) The authors describe how EAE develops spontaneously in C57BL/6 mice, but no markers for severity levels of EAE are found. Please provide any evidence of spontaneous EAE in this study's C57BL mice. The absence of EAE markers in CBA and BALB mice should be confirmed.
2) Because proteins are primary targets for IGs, it is possible that a subset of IGs could specifically target nucleases, allowing impure preparation to maintain RNAse activity. Please provide LC/MS proteomic analysis for the RNA hydrolysis specimens prepared from C57BL CBA and BALB mice.
3) Please provide some EMSA results to validate any significant physical IgG-labeled miRNA interactions?
4) The authors provide CBA and BALB control oligonucleotide hydrolysis in figure 2. Please provide additional controls from specimens prepared from MOG immunized corresponding mice. Controls related to C57BL mice are also lacking.
Author Response
Abzymes still not defined as an abbreviation. Antibodies-abzymes… see above.
Autoantibodies with catalytic activity are auto-abzymes. Always use the same wording. Many different "expressions" are in the manuscript: Antibodies-abzymes, Abs-abzymes, catalytic abzymes, auto-Abs-abzymes, Abzs, catalytic Abs-abzymes, IgGs-abzymes,
Answer: Sorry, but abzymes are not always auto-antibodies, they can also be developed against external antigens. In order to avoid tuftology in different phrases, depending on their meaning, it is necessary to use different modifications of the words antibodies-abzymes. Taking into account your remark, an attempt was made to unify the designation of the words antibodies-abzymes.
The definition of this term is given: Antibodies with catalytic activities (antibodies-enzymes; abzymes or Abzs) against transition states of various chemical reactions. Then some variants of the term were replaced.
Introduction: "AIDs development" is not correct: either AID development or development of AIDs.
Answer:
It was corrected
Figure 1: please avoid IgG1, IgG2, IgG3, as they are specified as IgG-subclasses. Instead, lable the curves with the specific antigen, e.g. before immunization, DNA/Histone, MOG.
Answer:
It was corrected
Figure 3: Still not clear which catalytic activity is shown after Time zero. Spontaneous EAE of non-immunized mice, MOG or DNA/histone immunized mice. If non-immunized mice are shown, the indication about immunizations is not needed. Otherwise show separate curves for spontaneous, MOG and DNA-Histone immunized mice.
Answer:
It was corrected
Discussion: "Somewhat unexpected were the results of a decrease in catalase activity in the period from 50 (kcat = 1.1x103 min-1) to three months of mice life (kcat = 40.7 min-1) by a factor of 27.4 (Figure 3)." Should be "from 50 days to three months…"
Answer:
It was corrected
"In the case of many antigens, including DNAs, RNAs, oligosaccharides, proteins, 301 lipids, etc.), a possible way of producing Abs…" the closing ")" should be removed.
Answer:
It was corrected
I am still missing the correlation of catalytic activity with the clinical score of EAE, or SLE respectively.
Answer: At the onset of the disease - up to about a year or even longer - mice do not have pronounced clinical signs of the disease. Taking into account your remark, the following text has been added to the article:
It was shown that C57BL/6 mice are characterized by a very slow spontaneous and MOG-induced development of EAE [3,4,17-20]. Some typical indicators of EAE development (optic neuritis and other clinical or histological evidence) appear in C57BL/6 mice only 1-2 years after spontaneous or MOG-accelerated evolution of EAE [3,4,17-20]. The appearance of auto-Abs hydrolyzing DNA, proteins and oligosaccharides was revealed as the earliest and statistically significant and undoubtedly important marker of the beginning of many autoimmune diseases in humans and mice prone to AIDs (for review, see [17-20]). Enzymatic activities of abzymes are veraciously detectable before the appearance of typical known medical and biochemical markers of different AIDs at the pre-disease stage [6-11,17-20]. At the pre-disease stage and onset of different AIDs, the concentrations of different auto-Abs usually correspond to the indices spans, which are typical for healthy humans and experimental mice. The emerging of abzymes may authentically testify about the beginning of AIDs, while the increase in their enzymatic activities is coupled with the development of deep pathologies [6-11,17-20]. In this work, we analyzed the changes in the catalase activity of antibodies at the early stages of the development of EAE in C57BL/6 mice.
Thanks for your comments
Best wishes
Prof Georgy Novinsky

This manuscript is a resubmission of an earlier submission. The following is a list of the peer review reports and author responses from that submission.
Round 1
Reviewer 1 Report
Even though I found the manuscript to be somewhat interesting, it contains numerous experimental flaws:
1) There are no experimental controls for Ig's RNAse activity with or without specific Ig's, and the reactions were carried out without the use of RNAse inhibitors. The authors claim that prepared Igs have no RNAse activity, but they provide no evidence.
2) It appears that a large portion of the data is missing. Control experiments with poliG, for example, are missing; the authors show hydrolysis activity only with a portion of Igs when performing reactions with poliA. Furthermore, the hydrolysis data is inconsistent.
3) Spatial miRNA structures and homooligonucleotide and miRNA hydrolysis are not correlated. As a result, the data indicate non-specific miRNA hydrolysis that could occur in buffer solution plus minimal RNAses present in sample preparations.
4) MiR155's affinity for Igs has no physiological significance.
5) I can't find any information about how the experiments depicted in figures 9 and 10 were carried out. Is there a change in the global RNAse enzymatic activity of blood plasma during animal immunization? Please provide qPCR analysis of selected miRNAs in blood plasma during immunization if no. If experiments were conducted with Igs prepared from animals at different immunization points, this indicates that there is some significant technical bias in the experiment with data normalization, and the data described is useless. So, how is this in vitro data physiologically relevant?
6) please provide RNA hydrolysis data with commercially available IGs.
Author Response
Even though I found the manuscript to be somewhat interesting, it contains numerous experimental flaws:
- There are no experimental controls for Ig's RNAse activity with or without specific Ig's, and the reactions were carried out without the use of RNAse inhibitors. The authors claim that prepared Igs have no RNAse activity, but they provide no evidence.
Answer: As we have shown in a number of studies, antibodies with RNase, DNase, and protease activities have the same amino acid residues in their active centers as the classical enzymes: RNases, DNases, and proteases [1-5]. Given this, the use of specific inhibitors of canonical enzymes leads to the inhibition of the activities of abzymes with the corresponding activities, and their use cannot help in the detection of classical enzyme impurities. The evidence that the antibody preparations do not contain canonical RNase impurities is shown in Figure 1. It can be seen that only IgGs with a molecular mass of 150 kDa have RNase activity. Canonical RNases have vastly lower molecular masses (13–15 kDa) than IgGs (150 kDa) and significantly higher activities in comparison with antibodies. In the case of the presence of microimpurities of classical RNases, activity peaks corresponding to nucleases with molecular weights of 10–40 kDa would first appear. Thus, the coincidence of the positions of RNase activity peak and protein band of 150 kDa IgGs directly indicates that mice IgGmix hydrolyze miRNA, and it is not contaminated with classical RNases.
References:
- Timofeeva, A. M., Buneva, V. N.,and Nevinsky, G. A. 2016. Systemic lupus erythematosus: molecular cloning and analysis of recombinant monoclonal kappa light chain NGTA1-Me-pro with two metalloprotease active centers. Molecular BioSystems 12:3556-3566.
- Timofeeva, A. M., Ivanisenko, N. V., Buneva, V.N., Nevinsky, G. A. 2015. Systemic lupus erythematosus: molecular cloning and analysis of recombinant monoclonal kappa light chain NGTA2-Me-pro-Tr possessing two different activities-trypsin-like and metalloprotease. InternationalImmunology 27:633-645.
- Timofeeva, A .M., Buneva, V. N.,and Nevinsky, G. A. SLE: Unusual Recombinant Monoclonal Light Chain NGTA3-Pro-DNase Possessing Three Different Activities Trypsin-like, Metalloprotease and DNase. Lupus Open Access 2017, 2:127.
- Bezuglova, A. V., Buneva, V. N., Nevinsky, G. A. 2011. Systemic lupus erythematosus: monoclonal light chains of immunoglobulins against myelin basic protein possess proteolytic and DNase activities. Russian Journal of Immunology 5:215-227.
- Nevinsky, G.A. The extreme diversity of autoantibodies and abzymes against different antigens in patients with various autoimmune diseases. Chapter in the book ”Advances in Medicine and Biology” Nova Science Publishers, Inc.; New York, USA, 2021, 184, 1-130.
- It appears that a large portion of the data is missing. Control experiments with poliG, for example, are missing; the authors show hydrolysis activity only with a portion of Igs when performing reactions with poliA. Furthermore, the hydrolysis data is inconsistent.
Answer: Sorry, but this remark is not entirely clear. All preparations from the blood of EAE mice hydrolyze ribooligonucleotides, but with different efficiency (Figure 2). They hydrolyze the worst (pA)23. But this is not surprising, since even canonical ribonucleases from some sources hydrolyze oligoadenylates worse. To compare the activity of different homooligonucleotides, the reaction mixtures were incubated under the same conditions - 2 hours. If mixtures with (pA)23 are incubated for 7–10 hours, then the patterns of hydrolysis of this adenylate become similar to those for (pC)23 and (pU)23, when all the stripes are clearly visible. But in the article, for comparison, all data are given corresponding to two hours. There are no control experiments with oligo(G), since these oligonucleotides form associates starting from 3-4 nucleotides, and it seems impossible to obtain homogeneous preparations (pG)23. Synthetics refuse to synthesize oligoG.
- Spatial miRNA structures and homooligonucleotide and miRNA hydrolysis are not correlated. As a result, the data indicate non-specific miRNA hydrolysis that could occur in buffer solution plus minimal RNAses present in sample preparations.
Answer: Sorry, but Figure 1 shows the absent of classical ribonucleases. According to a theoretical estimate, the human immune system is capable of synthesizing up to one million antibodies against the same antigen with very different properties. Moreover, according to our previously published data, the blood of, for example, patients with systemic lupus erythematosus contains about 1000-2000 antibodies against DNA and against MBP and some other proteins with very different properties. From 30 to 40% of all polyclonal antibodies to many antigens are abzymes that differ in the rate of hydrolysis of these antigens, optimal pH values, dependence and independence from metal ions, etc. In the blood of patients and animals with autoimmune diseases, not only antibodies against miRNAs, but also against homooligonucleotides are produced. At the same time, immunization of rabbits with any of the homo-RNAs (Cn, Un, An) leads to the production of abzymes that hydrolyze all homooligonucleotides, but with a different efficiency. Therefore, it is not possible to expect some good correlation in the hydrolysis of homooligonucleotides by antibodies against homooligos antigens, hetero-RNAs, and microRNAs. The relative amount of antibodies against different antigens depends on their concentration in the blood and specific ability of antigens to stimulate the formation of antibodies. Thus, antibodies against homosequences predominantly hydrolyze homooligonucleotides, and against microRNAs, these microRNAs.
Previously, we have not published data that a set of antibodies-abzymes from the blood of EAE mice similar to antibodies from the blood of patients with autoimmune diseases (revieved in [4-9]; article), are capable of hydrolyzing various RNAs, including homooligos and homopolynucleotides. With this in mind, we included in the article data on the hydrolysis of homooligonucleotides with IgGs of EAE mice. This is interesting, since the production of antibodies capable of hydrolyzing homooligonucleotides can also depend on the stage of pathology development.
- MiR155's affinity for Igs has no physiological significance.
Answer: Sorry, but there are very big problems with understanding the biological significance of biocatalysis. The Michaelis constant means the concentration of the substrate at which half of the maximum rate of the enzyme is reached. However, a very large number of enzymes have an affinity for a substrate that is an 10-100 orders of magnitude lower than the content of this substrate in cells and biological fluids. There are many such examples. For example, the content of ATP in human blood ~1-5 μM, while Km for ATP in the case of ATP-dependent creatine kinase is ~1 mM. However, the reaction does not proceed only when the substrate concentration is zero. But enzymes catalyze different reactions even at very low substrate - non-zero concentrations.
- I can't find any information about how the experiments depicted in figures 9 and 10 were carried out. Is there a change in the global RNAse enzymatic activity of blood plasma during animal immunization? Please provide qPCR analysis of selected miRNAs in blood plasma during immunization if no. If experiments were conducted with Igs prepared from animals at different immunization points, this indicates that there is some significant technical bias in the experiment with data normalization, and the data described is useless. So, how is this in vitro data physiologically relevant?
Answer: Analysis to assess the relative concentrations of miRNAs in the blood of mice is a separate and very laborious task. This work did not include such an analysis.
To calculate the average activity values for each of the groups of mice and substrates, hydrolysis patterns were used, some of which are shown in Figures 2-5. Considering your comment, the following information has been added to the “Statistical analysis” section.
Based on the data on the hydrolysis of each of the homooligonucleotides and micro-RNAs, an estimation of relative activity was made using the loss of each of the initial substrates after their incubation with each of the individual antibody preparation (some examples of substrates hydrolysis are shown in Figures 2-5). Then, the average value of antibody activity (mean ± standard deviation) was calculated using relative activities of seven IgG preparations of each of the analyzed groups of mice and shown in the Figures 9 and 10.
6) Please provide RNA hydrolysis data with commercially available IGs.
Answer: Sorry, but this doesn't make sense. Having been involved in abzymes study for many years, we tested commercially available preparations of IgGs from different companies. Unfortunately, they are poorly purified and contain microimpurities of various canonical enzymes, including ribonucleases. Such preparations are not suitable for the analysis of the catalytic activities of antibodies. Such commercial preparations hydrolyze DNA and RNA after SDS-PAGE in zones corresponding to 10 to 40 kDa, which are canonical DNases and RNases, but not antibodies. In our experiments (Figure 1), we show that there are no such impurities in our preparations from the blood of EAE mice. We have developed special antibody purification methods to remove all possible impurities, the preparations of which do not contain impurities of any enzymes, including canonical ribonucleases.
The introduction states that the blood of healthy donors and animals does not contain abzymes that hydrolyze RNA and DNA. We have shown that the blood of some healthy mice not prone to spontaneous development of autoimmune diseases, such as CBA, (CBAxC57BL)F1 and BALB mice, also does not contain abzymes. Based on your comment, we have conducted additional experiments using IgG antibodies purified by standard methods for obtaining preparations that do not contain impurities of any enzymes, similar to the isolation of antibodies from the blood plasma of EAE mice. It was shown that IgG antibodies from the blood of healthy CBA and BALB mice do not hydrolyze miRNA. This data is added to the Figures 2 and 3. These data indicate that the methods developed by us make it possible to isolate antibodies that do not contain impurities of canonical enzymes and that the presence of abzymes is a feature of people and animals with autoimmune diseases.
Thanks for the helpful notes
With best regards
Georgy A. Nevinsky
Reviewer 2 Report
The article by Andrey E. Urusov et. al. entitled " Experimental autoimmune encephalomyelitis of mice: IgGs from the sera of mice hydrolyze miRNAs " is an interesting study on systematic analysis of production of antibodies and abzymes possessing hydrolyzing miRNA activity with age. The authors found antibodies with RNase activity from EAE-prone C57BL/6 mice and ccompared the RNase activity of IgG obtained by spontaneous EAE or myelin oligodendrocyte glycoprotein (MOG) immunization. The spontaneous development of EAE leads not to an increase but to a permanent decrease of IgGs activity of hydrolysis of RNA-substrates. This can lead to a decrease in the production of antibodies and abzymes that hydrolyze miRNAs with age mice.
In general, the article is written clearly and presents interesting data, even if the figures and tables are not well represented and there are several typo and grammatical error in English. This article would be of interest to scientists who are focusing on the study of catalytic antibody and autoimmune disease.
I have the following comments and concerns.
1. Please describe the mechanism of how IgG hydrolyzes RNA.
2. Why were these 6 types of miRNAs used in the experiment? Please describe the relationship between EAE and these miRNAs.
3. Why did immunization with MOG increase RNA hydrolysis activity?
Page 1, line 40, “[13]” should be “[12]”. Please correct and check other references.
Page 9, line 295, “ABZs abzymes” should be “ABZs”, Please correct and check other abbreviations.
Page 11, Figure 10C, one of “miR219a-5p” should be “miR-146a-4p”. Please correct.
Too many self-citations in reference. Please also cite the papers of other researchers.
There are many careless grammatical and typographical errors throughout the manuscript. I would encourage the authors to proofread the manuscript carefully.
Author Response
The article by Andrey E. Urusov et. al. entitled " Experimental autoimmune encephalomyelitis of mice: IgGs from the sera of mice hydrolyze miRNAs " is an interesting study on systematic analysis of production of antibodies and abzymes possessing hydrolyzing miRNA activity with age. The authors found antibodies with RNase activity from EAE-prone C57BL/6 mice and ccompared the RNase activity of IgG obtained by spontaneous EAE or myelin oligodendrocyte glycoprotein (MOG) immunization. The spontaneous development of EAE leads not to an increase but to a permanent decrease of IgGs activity of hydrolysis of RNA-substrates. This can lead to a decrease in the production of antibodies and abzymes that hydrolyze miRNAs with age mice.
In general, the article is written clearly and presents interesting data, even if the figures and tables are not well represented and there are several typo and grammatical error in English. This article would be of interest to scientists who are focusing on the study of catalytic antibody and autoimmune disease.
I have the following comments and concerns.
- Please describe the mechanism of how IgG hydrolyzes RNA.
This question is currently difficult to answer, since we have not analyzed the DNA sequences and corresponding to them protein sequences of IgGs hydrolyzing microRNAs. However, we have shown in a number of studies that antibodies with DNase, and protease activities have the same amino acid residues in their active centers as the classical enzymes DNases and proteases [ ]. Given this, it should be assumed that the recognition sites of antibodies that hydrolyze RNAs and micro-RNAs also contain amino acid residues that are present in the active centers of canonical ribonucleases. These mechanisms associated with the action of two imidazole residues in the reaction of nucleophilic substitution associated with acid-base catalysis are well described in a number of articles and books on biocatalysis. Considering this, from our point of view, it is not productively to describe the known mechanisms of action of classical ribonucleases in this article.
References:
- Timofeeva, A. M., Buneva, V. N.,and Nevinsky, G. A. 2016. Systemic lupus erythematosus: molecular cloning and analysis of recombinant monoclonal kappa light chain NGTA1-Me-pro with two metalloprotease active centers. Molecular BioSystems 12:3556-3566.
- Timofeeva, A. M., Ivanisenko, N. V., Buneva, V.N., Nevinsky, G. A. 2015. Systemic lupus erythematosus: molecular cloning and analysis of recombinant monoclonal kappa light chain NGTA2-Me-pro-Tr possessing two different activities-trypsin-like and metalloprotease. InternationalImmunology 27:633-645.
- Timofeeva, A .M., Buneva, V. N.,and Nevinsky, G. A. SLE: Unusual Recombinant Monoclonal Light Chain NGTA3-Pro-DNase Possessing Three Different Activities Trypsin-like, Metalloprotease and DNase. Lupus Open Access 2017, 2:127.
- Bezuglova, A. V., Buneva, V. N., Nevinsky, G. A. 2011. Systemic lupus erythematosus: monoclonal light chains of immunoglobulins against myelin basic protein possess proteolytic and DNase activities. Russian Journal of Immunology 5:215-227.
- Nevinsky, G.A. The extreme diversity of autoantibodies and abzymes against different antigens in patients with various autoimmune diseases. Chapter in the book ”Advances in Medicine and Biology” Nova Science Publishers, Inc.; New York, USA, 2021, 184, 1-130.
- Why were these 6 types of miRNAs used in the experiment? Please describe the relationship between EAE and these miRNAs.
Answer: The C57BL/6 mice we used are a model of multiple sclerosis in humans. It should be assumed that, to some extent, the development of multiple sclerosis in humans and EАE in mice occurs in a similar way. As stated in the “Experimental Part”, we analyzed the hydrolysis of six miRNAs, two neuroregulatory and four immunoregulatory miRNAs characterized by impaired expression in patients with human multiple sclerosis [28]. This data is in short form added to the “Results”.
To analyze the sites of RNAs hydrolysis, we used three fluorescently 5’-labeled homo-oligonucleotides 5'-Flu-(pC)23, 5'-Flu-(pU)23, and 5'-Flu-(pA)23, as well as six 5'-Flu-miRNAs; two neuroregulatory miRNAs (miR-219a-5p, miR-9-5p) and four of immunoregulatory miRNAs (miR-21-3p, miR-146a-3p, miR-155-5p, and miR-326) characterized by impaired expression in patients with human multiple sclerosis [28].
- Why did immunization with MOG increase RNA hydrolysis activity?
Answer: According to the totality of our data, the development of autoimmune diseases begins with a violation of the differentiation profile of bone marrow stem cells [5-9]. At first glance, it could be assumed that immunization of mice with MOG and MBP should lead to such a violation of the profile, when lymphocytes appear that hydrolyze MOG and MBP, and treatment of mice with DNA to B-cells producing anti-DNA antibodies and DNase abzymes. However, with an immune response associated with a violation of the differentiation profile of stem cells, an expanded formation of lymphocytes occurs, producing antibodies and abzymes to various antigens. For example, when mice are immunized with MOG and MBP, cells are formed that produce IgGs hydrolyzing not only these antigens, but also DNA and histones, while immunization with DNA results in formation of Abs hydrolyzing DNA, MOG, MBP, and histones. The formation of an extended response, most likely, can also lead to the formation of lymphocytes that hydrolyze RNA and microRNA. So far, it is difficult to answer at this moment the question of the mechanisms of the extended response of the production of lymphocytes synthesizing antibodies to different autoantigens. It can be assumed that disruption of the Blood-brain barrier by one of the antigens may lead to the possibility of penetration of other antigens, including miRNAs and stimulate formation of Abs against micro-RNA. But at this stage, this can only be considered as a hypothesis.
Page 1, line 40, “[13]” should be “[12]”. Please correct and check other references.
Answer: It was corrected
Page 9, line 295, “ABZs abzymes” should be “ABZs”, Please correct and check other abbreviations.
Answer: It was corrected
Page 11, Figure 10C, one of “miR219a-5p” should be “miR-146a-3p”. Please correct.
Answer: It was corrected
Too many self-citations in reference. Please also cite the papers of other researchers.
Answer: Sorry, we always try to insert articles by other authors, but it doesn't work. There are practically no suitable articles in which there is data in this area of immunology. If you tell us which article of which author could be added according to the meaning of the article, we will definitely insert it.
There are many careless grammatical and typographical errors throughout the manuscript. I would encourage the authors to proofread the manuscript carefully.
Answer: It was corrected
Thanks for the helpful notes
With best regards
Georgy A. Nevinsky